# Implications of armed conflict for maternal and child health: A regression analysis of data from 181 countries for 2000–2019

Mohammed Jawad[1]*, Thomas Hone[1], Eszter P. Vamos[1], Valeria Cetorelli[2], Christopher Millett[1,3]

1 Public Health Policy Evaluation Unit, Imperial College London, London, United Kingdom, 2 United Nations Relief and Works Agency for Palestine Refugees in the Near East, Amman, Jordan, 3 Comprehensive Health Research Centre, NOVA National School of Public Health, NOVA University of Lisbon, Portugal

* mohammed.jawad06@imperial.ac.uk

## Abstract

### Background

Armed conflicts have major indirect health impacts in addition to the direct harms from violence. They create enduring political instability, destabilise health systems, and foster negative socioeconomic and environmental conditions—all of which constrain efforts to reduce maternal and child mortality. The detrimental impacts of conflict on global maternal and child health are not robustly quantified. This study assesses the association between conflict and maternal and child health globally.

### Methods and findings

Data for 181 countries (2000–2019) from the Uppsala Conflict Data Program and World Bank were analysed using panel regression models. Primary outcomes were maternal, under-5, infant, and neonatal mortality rates. Secondary outcomes were delivery by a skilled birth attendant and diphtheria, pertussis, and tetanus (DPT) and measles vaccination coverage. Models were adjusted for 10 confounders, country and year fixed effects, and conflict lagged by 1 year. Further lagged associations up to 10 years post-conflict were tested. The number of excess deaths due to conflict was estimated. Out of 3,718 country–year observations, 522 (14.0%) had minor conflicts and 148 (4.0%) had wars. In adjusted models, conflicts classified as wars were associated with an increase in maternal mortality of 36.9 maternal deaths per 100,000 live births (95% CI 1.9–72.0; 0.3 million excess deaths [95% CI 0.2 million–0.4 million] over the study period), an increase in infant mortality of 2.8 per 1,000 live births (95% CI 0.1–5.5; 2.0 million excess deaths [95% CI 1.6 million–2.5 million]), a decrease in DPT vaccination coverage of 4.9% (95% CI 1.5%–8.3%), and a decrease in measles vaccination coverage of 7.3% (95% CI 2.7%–11.8%). The long-term impacts of war were demonstrated by associated increases in maternal mortality observed for up to 7 years, in under-5 mortality for 3–5 years, in infant mortality for up to 8 years, in DPT vaccination coverage for up to 3 years, and in measles vaccination coverage for up to 2 years. No evidence of association between armed conflict and neonatal mortality or delivery by a

---

**Data Availability Statement:** ll primary and secondary outcome data (maternal mortality, under-5 mortality, infant mortality, neonatal mortality, skilled birth attendance, measles

vaccination, DTP vaccination) and selected covariates (GDP per capita and age dependency ratio) are available from the World Bank at https://bit.ly/2yUZmdS. Covariates (population density, urban residence, male education, temperature and rainfall) are available from the IHME at https://bit.ly/2U3LQEc. The covariates earthquakes and droughts are available from the International Disaster Database at https://bit.ly/39f5zo9 The covariate Ethnic Fractionalisation Index is available from https://bit.ly/3iuLEan. The covariate Electoral Democracy Index is available from https://bit.ly/3IlflXI. Armed conflict data can be found from the UCDP at https://bit.ly/3iqR8CZ and from the MEPV at https://bit.ly/37reDWh.

**Funding:** MJ is supported by the Medical Research Council Doctoral Training Partnership. PHPE is supported by the NIHR School of Public Health Research. The funders had no role in study design, data collection and analysis, decision to publish, or preparation of the manuscript.

**Competing interests:** The authors have declared that no competing interests exist.

**Abbreviations:** DHS, Demographic and Health Surveys; DPT, diphtheria, pertussis, and tetanus; GDP, gross domestic product; LMICs, low- and middle-income countries; OECD, Organisation for Economic Co-operation and Development; UCDP, Uppsala Conflict Data Program; UCDP GED, Uppsala Conflict Data Program Georeferenced Event Dataset; SD, standard deviation; VIF, variance inflation factor.

skilled birth attendant was found. Study limitations include the ecological study design, which may mask sub-national variation in conflict intensity, and the quality of the underlying data.

## Conclusions

Our analysis indicates that armed conflict is associated with substantial and persistent excess maternal and child deaths globally, and with reductions in key measures that indicate reduced availability of organised healthcare. These findings highlight the importance of protecting women and children from the indirect harms of conflict, including those relating to health system deterioration and worsening socioeconomic conditions.

## Author summary

### Why was this study done?

- There is little and inconsistent evidence on the maternal and child health impacts of armed conflict.

- Furthermore, there is a poor understanding of how armed conflict affects maternal and child health in the medium to long term.

### What did the researchers do and find?

- We modelled the association between armed conflicts and maternal and child mortality, deliveries by a skilled birth attendant, and vaccination coverage for measles and diphtheria, pertussis, and tetanus (DPT). We used panel regression models that included 181 countries. We used the models to calculate the number of excess maternal and child deaths that can be attributed to armed conflict.

- Our models indicate that over the period 2000–2019, armed conflicts classified as wars were associated with an increase of 36.9 maternal deaths per 100,000 live births, an increase of 2.8 infant (under 1 year old) deaths per 1,000 live births, 7.3% lower measles vaccination coverage, and 4.9% lower DPT coverage. This approximated to 300,000 avoidable maternal deaths and 2 million avoidable infant deaths over the study period.

- We were able to show that these associations persisted for many years following the onset of war, including up to 7 years for maternal mortality and up to 8 years for infant mortality.

- A range of sensitivity analyses show that even under different model assumptions and using different data sources, the association between war and maternal and child deaths continued to be high.

### What do these findings mean?

- Our study suggests that armed conflicts may result in a substantial number of preventable maternal and child deaths globally and that mortality rate increases may persist for

years following the onset of these conflicts. This is likely to threaten achievement of the Sustainable Development Goals related to maternal and child health.

- We highlight the importance of preserving functioning health systems so that women and children can be protected during and after armed conflicts.

## Introduction

Despite improvements globally, the burden of maternal and child mortality continues to be unacceptably high. There were an estimated 295,000 maternal deaths in 2017, of which 94% occurred in low- and middle-income countries (LMICs) [1]. Similarly, most of the global burden of under-5, infant, and neonatal mortality is in LMICs. Over 5 million children under the age of 5 years (of whom 75% are under the age of 1 year) and 2.6 million aged under 28 days died in 2018 [2].

Achieving the Sustainable Development Goal targets for maternal (less than 70 per 100,000 live births), under-5 (less than 25 per 1,000 live births), and neonatal mortality (less than 12 per 1,000 live births) [3] in conflict-affected countries is a major challenge. Conflicts exacerbate political instability, destabilise social and welfare systems including health systems, interrupt water and food supplies, and increase poverty, unemployment, and homelessness—all of which negatively affect maternal and child health [4]. Estimates from 2017 indicate that approximately 420 million children younger than 18 years were living in areas affected by conflict [5], and 630 million women and children were living dangerously close (<50 km) to conflicts [6].

Despite growing recognition of the detrimental effects of conflict on maternal and child health, the literature is bereft of studies robustly quantifying the association between conflict and maternal and child health, with many inconsistent findings [7–13]. Prior research often failed to adequately define conflict or to adequately define data sources [7,8,13], focused on sub-Saharan Africa or LMICs only [7,8,10–15], had limitations in modelling approaches [7–11], evaluated only short-term associations [7–9,11–15], or assessed single mortality indicators [7–11,14,15]. For example, one major critique of the conflict literature is its approach to confounding variables. A review by Dixon of 46 quantitative studies found that over 200 independent variables had been assessed as possible causes of war, with a high degree of consensus in no more than 4 (demographics, economy, history of insecurity, and democratisation) [16]. Many independent variables may be collinear with armed conflict, which highlights the importance of grounding conflict models in conceptual frameworks and measuring collinearity between variables.

The *Lancet* Series on Women's and Children's Health in Conflict Settings lamented the lack of systematic evidence and readily available data in this field [17]. A robust, comprehensive, global assessment of the impacts of conflict on maternal and children's health is lacking. This study assesses global associations between conflict and maternal and child mortality and the use of health services that are known to reduce mortality (skilled birth delivery and vaccination coverage of diphtheria, pertussis, and tetanus [DPT] and measles), in both the short and long term.

## Methods

### Design and setting

This panel (longitudinal) study used data from 181 countries for the years 2000 to 2019. Country was the unit of analysis. It explored the relationships between countries' conflict

status and maternal and child health outcomes. Panel regression modelling approaches are frequently used in global health research and country-level analyses over time [18]. They allow changes in health outcomes to be associated with changes in variables such as exposure to conflict over time, whilst adjusting for potential confounders, including country fixed effects and time trends. Although a prospectively written plan is not available, the analysis plan was based on previous conflict and health analytical frameworks developed by the study authors, which included prespecified model structures, covariate selection, and time lags [19]. We used all available maternal and child health outcomes available, to minimise reporting biases. The study followed RECORD reporting guidelines (S1 RECORD Checklist).

## Data

One data source in this study was the Uppsala Conflict Data Program (UCDP) Georeferenced Event Dataset (UCDP GED) [20]. The UCDP has dedicated research teams that systematically determine the presence of conflict using definitions that are consistent and comparable across countries and years [20]. The UCDP GED uses 'events' as the unit of analysis, defined as 'the use of armed force by an organised actor against another organised actor or civilians, resulting in at least one direct death for that location and time' [20]. An armed conflict is present when events cumulatively result in 25 or more battle-related deaths within in calendar–conflict year. The UCDP GED has been used in peer-reviewed publications previously [8,10–12], and more details on its methodology can be found elsewhere [20]. The UCDP GED was aggregated into country–year observations for the analysis. Data for Palestine and Israel were omitted as the UCDP does not distinguish between them, which is problematic for attributing associated health outcomes for these states separately.

Country-level health data were obtained from the World Bank [21]. Specifically, these were rates of maternal, under-5, infant (<1 year), and neonatal (<28 days) mortality; delivery by a skilled birth attendant; and vaccination coverage of DPT and measles among 12- to 23-month-old children. The World Bank provides maternal and child health data from the UN Maternal Mortality Estimation Inter-agency Group and the UN Inter-agency Group for Child Mortality Estimation, whose task has been to improve data quality and produce yearly estimates for each country [22]. Data are primarily taken from vital registration systems (adjusting for underestimation); however, absent or incomplete data are modelled using multilevel regressions with social indicators (e.g., fertility rates, educational attainment, and health service coverage) [22]. The resulting mortality estimates are comparable across countries and years. The latest maternal mortality data were available from 2000 to 2017, and child mortality data from 2000 to 2019 [21]. Data were fully complete for all primary outcomes, while data were missing for 52.6% of country–year observations for skilled birth attendance and 0.5% of observations for both DPT and measles vaccination coverage. For delivery by a skilled birth attendant, data were not missing at random: The number of countries with data for this variable varied from 10 to 135 countries per year, with fewer after 2015. Country–year observations during armed conflict constituted 20.6% ($n = 443/2,148$) of missing data but 13.4% ($n = 227/1,692$) of non-missing data, suggesting selection bias should be considered when interpreting these findings.

Covariate data were taken from several sources including the World Bank [21], Institute for Health Metrics and Evaluation [23], International Disaster Database [24], V-Dem Institute [25], and Historical Index of Ethnic Fractionalization Dataset [26] (see below for descriptions of these variables).

## Measures

Primary outcome measures were maternal, under-5, infant, and neonatal mortality rates. The maternal mortality ratio was the number of women who died from pregnancy-related causes while pregnant or within 42 days of pregnancy termination per 100,000 live births. Under-5 mortality was the probability per 1,000 that a newborn baby died before reaching age 5 years. The infant and neonatal mortality rates were the number dying before reaching 1 year of age and 28 days of age, respectively, per 1,000 live births.

Secondary outcome measures were delivery by a skilled birth attendant and vaccination coverage of DPT and measles among 12- to 23-month-old children. Delivery by a skilled birth attendant was the percentage of deliveries attended by personnel trained to give the necessary supervision, care, and advice to women during pregnancy, labour, and the postpartum period; to conduct deliveries on their own; and to care for newborns [27]. DPT and measles vaccinations were chosen instead of other vaccinations (e.g., PCV3, HIB3, rotavirus) as they were implemented in every country over the entire study period, making them comparable markers of health system performance over time. These secondary outcome measures were used to determine possible underlying mechanisms of adverse outcomes as they have known benefits for maternal and child health [28] and serve as proxies for the wider health system.

The exposure variable of interest was armed conflict, as defined by the UCDP. Due to the complexity of measuring conflict, 4 specifications were explored in the analyses. First, the most commonly employed specification of conflict in the literature is a binary variable indicating the presence of conflict for each country–year observation (0 = no, <25 battle-related deaths; 1 = yes, ≥25 battle-related deaths) [29]. Second, to better account for conflict intensity, this specification was expanded into a categorical variable with recommended cutoffs from the UCDP (0 = no, <25 battle-related deaths; 1 = minor conflict, 25–999 battle-related deaths; 2 = war, ≥1,000 battle-related deaths) [29]. This categorisation was made for each conflict per country–year observation, meaning that countries with many minor conflicts summing to 1,000 or more battle-related deaths per year were not categorised as war-affected. The rationale behind this decision was to prevent bias where large countries with many minor conflicts, such as India, were incorrectly classified as war-affected. This decision meant that 68 country–year observations were categorised as experiencing minor conflicts despite having 1,000 or more battle-related deaths; we found no difference in models that categorised these 68 country–year observations as minor conflict or war, so we opted for the former. Third, to account for the scale of battle-related deaths relative to the population, battle-related deaths per 100,000 population was used as a continuous variable. Fourth, this continuous measure was expressed as quintiles to model a non-linear relationship between conflict and mortality [10]. It should be noted that the 'no conflict' category for the binary and categorical specifications referred to fewer than 25 battle-related deaths, whereas for the quintile specification it referred to no battle-related deaths.

To adjust regression models for potential confounders, covariates were selected based on conceptual frameworks that outlined the theoretical causal pathways between conflict and health [30–32]. Ten suitable covariates were identified that are probable confounders and were included in the regression models—all expressed at the country–year level (see S1 Table for full definitions). To capture changes in national wealth, gross domestic product (GDP) per capita (current US dollars) [21] and Organisation for Economic Co-operation and Development (OECD) membership [33] were used. To capture changes in the political system, the Electoral Democracy Index was employed [25]. To adjust for changes in population demographics, population density [23], urban residence [23], and the age dependency ratio [21] were all included in models. To account for changes in population education, the mean

number of years of educational attainment per capita was included in the models [23]. To capture changes in ethnic group composition, the Historical Index of Ethnic Fractionalisation Dataset was used [26]. Finally, to capture changes and shocks in climate-related factors, indicators for the presence of droughts [24] and earthquakes [24], and the population-weighted mean temperature and rainfall, were used [23]. There were missing data in 4 covariates: GDP per capita (2.7%), age dependency ratio (3.9%), Electoral Democracy Index (11.8%), and Ethnic Fractionalisation Index (46.0%). The latter 2 were omitted from models due to high levels of missing data, but reintroduced in sensitivity analyses (see below). There were missing data in the skilled birth attendant outcome measure (55.9%), but it was included in the analysis to minimise reporting bias.

## Statistical analysis

All variables were descriptively analysed to produce means and standard deviations (SDs) of outcome measures by the 4 conflict specifications. The data were analysed using panel regression models to assess changes in exposure and outcomes over time and the associations between these changes. Fixed effects specifications were employed to adjust for all unobserved and observed time-invariant factors that could bias the analyses. Fixed effects specifications are frequently employed to robustly model dynamic and transitional associations over time. Regression models were based on the following equation:

$$\text{Outcome}_{it} = \beta_0 + \beta_1 \text{ Conflict}_{i(t-1)} + \beta_2 \text{ Covariates}_{it} + i + t + u_{it}$$

where $i$ is the country, $t$ is the year, and $u$ is the error term representing unobserved time-variant effects. Use of fixed effects specifications was supported by the Hausman test, which examines differences in coefficients between fixed and random effects specifications. We assessed model fit graphically and using post-regression diagnostics (e.g., variance inflation factor [VIF], studentized residuals, stem and leaf plots, Cook's D, and difference in fits) to identify collinear variables and outliers. We found no collinearity in our models (VIF < 2). Four observations (county–years Rwanda 1994, Bosnia and Herzegovina 1995, Congo 1997, and Eritrea 1999) were considered outliers due to having high leverage and influence, and were omitted from the model that specified conflict as a continuous variable. The variables of male and female education were collinear, and the latter was omitted due to poorer model fit.

Our main analyses were composed of 4 regression models for each primary and secondary outcome, representing the 4 specifications of conflict. The variable of interest, conflict, was expressed as a lagged term (i.e., conflict status in the year before) to address potential reverse causation and account for the likely delay in any associated changes in health outcomes. Each model was adjusted for 10 covariates (GDP per capita, OECD member, population density, urban residence, age dependency ratio, male education, temperature, rainfall, earthquakes, and droughts), and year and country fixed effects. Country-clustered Huber–White robust standard errors were used to account for possible heteroscedasticity and serial correlation. The beta coefficients for each conflict variable are interpreted as the absolute change in the outcome associated with a 1-unit change in armed conflict status, which can be changing from no conflict to conflict (binary) or increasing intensity of conflict (categorical or continuous specifications). For the primary outcomes, the coefficients were used to calculate a relative percentage change and the number of predicted excess deaths using post-regression estimations. To assess longer-term associations, we modelled conflict as a variable lagged from 1 to 10 years, with separate models for each year of lag.

All analyses were conducted in Stata 15.

## Sensitivity analyses

The robustness of findings was tested using alternative model specifications including the sequential addition of covariates, the addition of 2 covariates that contained high rates of missing data (Electoral Democracy Index and Ethnic Fractionalisation Index), and random effects specifications. Alternative data sources for measuring armed conflict were explored and derived from the Major Episodes of Political Violence dataset compiled by the Center for Systemic Peace [34]. This dataset captures major conflicts globally and its binary conflict variable (0 = no conflict, 1 = conflict) is 97.6% specific and 97.7% sensitive to the UCDP 'war' binary variable (0 = no conflict, 1 = war).

## Results

Armed conflict was present in 670 of the 3,718 (18.0%) country–year observations, three-quarters of which were minor conflicts (n = 522, 77.9%). Higher maternal and child mortality rates and poorer health system performance was correlated with greater conflict intensity in descriptive analyses (Table 1).

## Maternal mortality

Conflict-affected countries had double the mean maternal mortality ratio of conflict-free countries (396.3 versus 169.3/100,000 live births), with the highest ratio in countries classified

**Table 1. Description of data used in the study.**

| Variable | Countries (n) | Observations (n) | Primary outcomes | | | | Secondary outcomes | | |
|---|---|---|---|---|---|---|---|---|---|
| | | | Maternal mortality ratio per 100,000 live births | Under-5 mortality rate per 1,000 live births | Infant mortality rate per 1,000 live births | Neonatal mortality rate per 1,000 live births | Births attended by skilled health staff (%) | DPT vaccination coverage (%) | Measles vaccination coverage (%) |
| **Total** | 186 | 3,718 | 210.6 (303.8) | 40.5 (42.3) | 28.9 (26.4) | 16.1 (12.9) | 88.6 (20.5) | 86.6 (15.1) | 85.8 (15.1) |
| **Armed conflict exposure** | | | | | | | | | |
| No[1] | 175 | 3,048 | 169.3 (262.4) | 34.6 (38.2) | 25.1 (24.0) | 14.1 (11.7) | 91.6 (16.9) | 89.2 (12.1) | 88.2 (12.6) |
| Yes | 86 | 670 | 396.3 (394.6) | 67.3 (49.3) | 46.3 (29.6) | 25.3 (13.8) | 69.7 (29.6) | 75.1 (20.7) | 75.2 (19.8) |
| Minor conflict[2] | 84 | 522 | 373.5 (387.1) | 65.2 (49.2) | 44.7 (29.3) | 24.2 (13.3) | 71.5 (29.1) | 77.8 (20.1) | 77.6 (19.7) |
| War[3] | 31 | 148 | 475.4 (411.2) | 74.9 (49.2) | 52.0 (29.9) | 29.3 (14.8) | 61.6 (31.0) | 65.3 (19.9) | 66.8 (18.0) |
| **Quintile of armed conflict exposure** | | | | | | | | | |
| None[4] | 166 | 2,718 | 154.1 (250.3) | 32.2 (36.8) | 23.5 (23.3) | 13.3 (11.4) | 92.8 (15.2) | 90.0 (11.4) | 88.9 (11.9) |
| First | 76 | 226 | 309.9 (326.6) | 55.5 (43.8) | 38.5 (26.6) | 20.8 (12.5) | 75.2 (26.7) | 82.6 (14.9) | 80.8 (15.8) |
| Second | 69 | 187 | 263.5 (287.6) | 52.1 (42.7) | 36.5 (25.1) | 20.4 (11.7) | 82.0 (23.1) | 84.0 (15.9) | 83.5 (17.0) |
| Third | 61 | 198 | 342.0 (388.0) | 59.7 (49.7) | 40.9 (29.8) | 22.3 (13.5) | 71.9 (30.8) | 81.4 (19.4) | 80.9 (18.7) |
| Fourth | 43 | 193 | 431.6 (422.5) | 69.7 (50.0) | 47.9 (29.8) | 26.1 (13.6) | 67.4 (30.5) | 74.4 (20.6) | 74.6 (20.4) |
| Fifth | 36 | 196 | 479.0 (400.3) | 78.0 (48.7) | 53.8 (29.5) | 29.6 (14.1) | 62.6 (29.5) | 65.7 (19.8) | 66.9 (18.0) |

Data given as mean (standard deviation).

[1]No conflict: <25 battle-related deaths per country–conflict–year.

[2]Minor conflict: 25–999 battle-related deaths per country–conflict–year.

[3]War: ≥1,000 battle-related deaths per country–conflict–year.

[4]Zero battle-related deaths per 100,000 population per country–year.

DPT, diphtheria, pertussis, and tetanus.

as being at war (475.4; Table 1). In adjusted regression models (Table 2), conflict (binary) was associated with an average absolute increase in maternal mortality of 22.5 per 100,000 live births (95% CI 1.7–43.2), which is a relative increase of 13.3% (95% CI 1.1%–25.4%) over the maternal mortality rate in conflict-free countries. Conflicts classified as wars were associated with an absolute increase in maternal mortality of 36.9 per 100,000 live births (95% CI 1.9–72.0) and a relative increase of 21.8% (95% CI 1.1%–42.6%). Wars were estimated to have caused 0.3 million (95% CI 0.2 million–0.4 million) excess maternal deaths over the study period. Increases in battle-related deaths were positively, but non-significantly, associated with an increase in maternal mortality (S2 Table). The third and fifth quintiles of conflict exposure were significantly associated with increased maternal mortality—by 32.1 (95% CI 5.7–58.5) and 55.8 per 100,000 live births (95% CI 16.3–95.3), respectively (S2 Table). War was significantly associated with increases in maternal mortality for up to 7 years following the onset of war (Fig 1).

### Under-5 mortality

The mean under-5 mortality rate was twice as high in conflict-affected (67.3/1,000 live births) as in conflict-free countries (34.6/1,000 live births), with the highest mortality rate in countries with conflicts classified as wars (74.9/1,000 live births; Table 1). In adjusted regression models, none of the 4 specifications of conflict was statistically associated with under-5 mortality. However, lagged models showed that war was significantly associated with higher under-5 mortality 3–8 years since the onset of war (Fig 1). Post-regression models estimated 3.2 million (95% CI 2.5 million–3.9 million) under-5 deaths associated with wars during the study period.

### Infant mortality

The mean infant mortality rate was higher in conflict-affected (46.3 per 1,000 live births) than conflict-free countries (25.1 per 1,000 live births), with the rate highest in countries with conflicts classified as wars (52.0 per 1,000 live births; Table 1). In adjusted regression models (Table 2), conflicts classified as wars were associated with an increase of 2.8 infant deaths per 1,000 live births (95% CI 0.1–5.9; Table 2). The fifth quintile of conflict intensity was associated with an increase in infant mortality of 3.8 per 1,000 live births (95% CI 0.2–7.3), and each battle-related death per 100,000 population was association with an increase of 0.1 per 1,000 live births (95% CI 0.0–0.2; S2 Table). Wars were associated with higher infant mortality for up to 8 years following the onset of war (Fig 1). Post-regression prediction models estimated 2.0 million (95% CI 1.6 million–2.5 million) infant deaths were due to wars during the study period.

### Neonatal mortality

The mean neonatal mortality rate was higher in conflict-affected (25.3 per 1,000 live births) compared with conflict-free (14.1 per 1,000 live births) countries, and highest in countries experiencing war (29.3 per 1,000 live births; Table 1). In all adjusted regression models (Table 2), including lagged analyses, conflict in all specifications was positively, although not significantly, associated with higher neonatal mortality, and the relationship was suggestive of a dose response.

### Skilled birth attendants and childhood vaccinations

Conflict-affected countries had 21.9% fewer births attended by skilled health staff, 14.1% lower DPT vaccination coverage, and 13.0% lower measles vaccination coverage than conflict-free countries (Table 1). These differences were approximately 10 percentage points larger for

**Table 2. The association between armed conflict and maternal and child mortality (adjusted beta coefficients, 95% confidence intervals).**

| Variable | Maternal mortality ratio per 100,000 live births | | Under-5 mortality rate per 1,000 live births | | Infant mortality rate per 1,000 live births | | Neonatal mortality rate per 1,000 live births | |
|---|---|---|---|---|---|---|---|---|
| | Binary conflict specification | Three-category conflict specification | Binary conflict specification | Three-category conflict specification | Binary conflict specification | Three-category conflict specification | Binary conflict specification | Three-category conflict specification |
| **Armed conflict —binary** | | | | | | | | |
| No | 0.00 (0.00, 0.00) | | 0.00 (0.00, 0.00) | | 0.00 (0.00, 0.00) | | 0.00 (0.00, 0.00) | |
| Yes | 22.48* (1.74, 43.22) | | 1.94 (−2.01, 5.88) | | 1.31 (−0.46, 3.08) | | 0.17 (−0.41, 0.75) | |
| **Armed conflict —3 categories** | | | | | | | | |
| No | | 0.00 (0.00, 0.00) | | 0.00 (0.00, 0.00) | | 0.00 (0.00, 0.00) | | 0.00 (0.00, 0.00) |
| Minor conflict | | 20.62 (−0.05, 41.28) | | 1.57 (−2.36, 5.50) | | 1.12 (−0.62, 2.85) | | 0.11 (−0.47, 0.69) |
| War | | 36.93* (1.86, 72.01) | | 4.76 (−0.52, 10.04) | | 2.79* (0.09, 5.48) | | 0.65 (−0.35, 1.65) |
| **Covariates** | | | | | | | | |
| GDP per capita | 2.89*** (1.62, 4.17) | 2.89*** (1.61, 4.16) | 0.63*** (0.39, 0.87) | 0.63*** (0.39, 0.87) | 0.33*** (0.22, 0.45) | 0.33*** (0.22, 0.45) | 0.11*** (0.07, 0.15) | 0.11*** (0.07, 0.15) |
| OECD member | 18.52 (−3.54, 40.58) | 18.42 (−3.62, 40.45) | 3.87 (−2.32, 10.06) | 3.85 (−2.33, 10.03) | 1.55 (−2.31, 5.41) | 1.54 (−2.31, 5.40) | 0.42 (−1.38, 2.23) | 0.42 (−1.38, 2.23) |
| Population density | −57.16 (−278.97, 164.64) | −56.44 (−276.66, 163.78) | −39.44 (−102.55, 23.66) | −39.23 (−102.05, 23.59) | −21.33 (−56.11, 13.45) | −21.21 (−55.84, 13.41) | −6.38 (−18.02, 5.26) | −6.35 (−17.93, 5.23) |
| Urban residence | −6.76 (−14.15, 0.63) | −6.69 (−13.99, 0.62) | −1.48* (−2.95, −0.01) | −1.47* (−2.92, −0.01) | −0.88* (−1.68, −0.08) | −0.87* (−1.66, −0.08) | −0.35* (−0.65, −0.05) | −0.34* (−0.64, −0.05) |
| Age dependency ratio | 0.40 (−1.49, 2.28) | 0.40 (−1.48, 2.29) | 0.15 (−0.16, 0.46) | 0.15 (−0.16, 0.46) | 0.21* (0.04, 0.37) | 0.21* (0.04, 0.37) | 0.13*** (0.06, 0.20) | 0.13*** (0.06, 0.20) |
| Male education | −41.79* (−77.83, −5.75) | −42.18* (−78.02, −6.34) | −4.19 (−10.43, 2.05) | −4.23 (−10.45, 1.99) | −1.49 (−4.85, 1.87) | −1.51 (−4.87, 1.84) | 0.24 (−1.15, 1.62) | 0.23 (−1.15, 1.61) |
| Temperature | 10.73** (4.23, 17.23) | 10.69** (4.22, 17.16) | 2.40*** (0.99, 3.81) | 2.39*** (0.99, 3.79) | 1.07** (0.34, 1.80) | 1.06** (0.34, 1.79) | 0.25 (−0.03, 0.53) | 0.24 (−0.03, 0.52) |
| Rainfall | −4.34 (−13.80, 5.12) | −4.35 (−13.78, 5.08) | 0.55 (−1.20, 2.30) | 0.53 (−1.20, 2.27) | 0.34 (−0.62, 1.31) | 0.33 (−0.62, 1.29) | 0.01 (−0.38, 0.40) | 0.01 (−0.38, 0.39) |
| Earthquakes | 5.78 (−2.24, 13.81) | 5.89 (−2.10, 13.88) | 1.53 (−0.46, 3.52) | 1.56 (−0.42, 3.55) | 0.60 (−0.14, 1.34) | 0.62 (−0.12, 1.36) | 0.38* (0.08, 0.68) | 0.39* (0.08, 0.69) |
| Droughts | 4.70 (−2.13, 11.53) | 4.73 (−2.10, 11.56) | 1.75** (0.60, 2.90) | 1.77** (0.62, 2.92) | 0.99** (0.37, 1.61) | 1.00** (0.38, 1.62) | 0.40** (0.12, 0.67) | 0.40** (0.12, 0.68) |
| Observations | 3,045 | 3,045 | 3,376 | 3,376 | 3,376 | 3,376 | 3,376 | 3,376 |
| Countries | 181 | 181 | 180 | 180 | 180 | 180 | 180 | 180 |

*$p < 0.05$,

**$p < 0.01$,

***$p < 0.001$. Data given as adjusted beta coefficient (95% confidence interval). Robust standard errors were employed. Each column is the output from 1 panel regression with fixed effects adjusted for the covariates in the table in addition to year dummies (not shown). Coefficients are interpreted as the absolute change in the dependent variable following a 1-unit change of the independent variable. GDP per capita is in current US dollars, and its unit is scaled up by 1,000. Population density represents the percentage of the population living in a density of >1,000 people/km$^2$. Urban residence represents the percentage of the population living in urban areas. The age dependency ratio represents the proportion of dependents (aged under 15 years or over 64 years) per 100 working-age individuals. Male education is expressed as years per capita and is age-standardised. Temperature is in degrees Celsius and is the mean population-weighted annual temperature. Rainfall is the mean population-weighted annual rainfall in millimetres per year, scaled down by 1,000. Earthquake and drought are binary variables representing their absence or presence. All armed conflict variables were lagged by 1 year.

GDP, gross domestic product; OECD, Organisation for Economic Co-operation and Development.

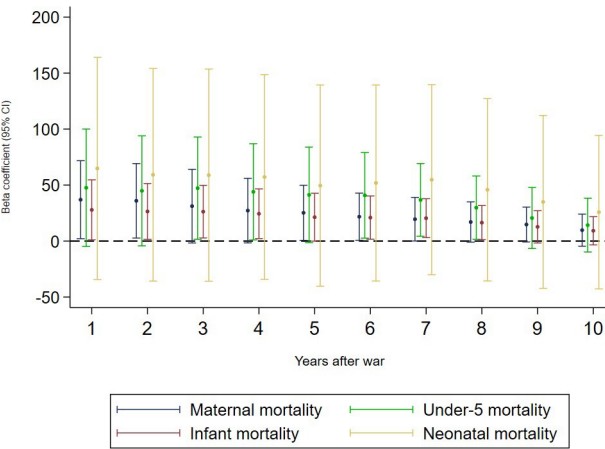

**Fig 1. The association between war and maternal and child mortality with up to 10-year lags.** Each line represents the output from a fixed effects panel regression model adjusted for 10 covariates and year fixed effects, showing the war coefficient only. Under-5 and infant mortality are scaled up by a factor of 10, and neonatal mortality is scaled up by a factor of 100, to fit the axis.

conflicts categorised as wars (Table 1). In adjusted regression models (Table 3), no association was found between any armed conflict specification and births attended by skilled health staff, including in lagged models. Armed conflict was associated with a reduction in measles vaccination coverage of 2.6% (95% CI 0.2%–5.0%), whilst war was associated with 4.9% (95% CI 1.5%–8.3%) and 7.3% (95% CI 2.7%–11.8%) reductions in DPT and measles vaccination coverage, respectively. There were reductions in vaccination coverage associated with the fifth quintile of conflict intensity and with battle-related deaths as a continuous measure (S3 Table). Wars were associated with reduced DPT vaccination coverage for 3 years, and reduced measles vaccination coverage for 2 years, after war onset (Fig 2).

## Sensitivity analyses

Sensitivity analyses demonstrated robustness of findings across all outcomes, and we present these data for maternal mortality. Within the sensitivity analyses, random effects models showed positive associations between armed conflict and maternal mortality across all 4 conflict specifications, consistent with those from fixed effects specifications (S4 Table). The sequential addition of covariates did not substantially alter armed conflict coefficients, suggesting stability of the models (S5 Table). The addition of the 2 covariates Electoral Democracy Index and Ethnic Fractionalisation Index had minimal impact on the coefficient for war (β 46.0, 95% CI 9.5–82.4; S5 Table, Model 11). Replacing the UCDP data with a binary armed conflict variable from the Major Episodes of Political Violence dataset also produced a similar and statistically significant coefficient (β 43.2, 95% CI 1.4–85.0; S6 Table).

## Discussion

This study uses robust quasi-experimental methods and comprehensive global datasets to demonstrate that conflict is associated with substantial increases in maternal, under-5, and infant mortality, in addition to decreases in vaccination coverage. There is evidence of a dose–response relationship, with increasing intensity of conflict, particularly war, associated with greater increases in mortality and reductions in vaccinations. We provide powerful and new global estimates of the associations between conflict and health that are beyond the limited

**Table 3. The association between armed conflict and maternal and child health indicators (adjusted beta coefficients, 95% confidence intervals).**

| Variable | Births attended by skilled health staff | | DPT immunisation | | Measles immunisation | |
|---|---|---|---|---|---|---|
| | Binary conflict specification | Three-category conflict specification | Binary conflict specification | Three-category conflict specification | Binary conflict specification | Three-category conflict specification |
| **Armed conflict—binary** | | | | | | |
| No | 0.00 (0.00, 0.00) | | 0.00 (0.00, 0.00) | | 0.00 (0.00, 0.00) | |
| Yes | −0.83 (−2.84, 1.19) | | −1.47 (−3.45, 0.52) | | −2.62* (−5.05, −0.20) | |
| **Armed conflict—3 categories** | | | | | | |
| No | | 0.00 (0.00, 0.00) | | 0.00 (0.00, 0.00) | | 0.00 (0.00, 0.00) |
| Minor conflict | | −0.97 (−2.99, 1.06) | | −1.02 (−2.96, 0.91) | | −2.03 (−4.27, 0.21) |
| War | | 0.71 (−3.24, 4.66) | | −4.91** (−8.32, −1.49) | | −7.25** (−11.81, −2.70) |
| **Covariates** | | | | | | |
| GDP per capita | −0.09** (−0.15, −0.02) | −0.09** (−0.15, −0.02) | −0.09 (−0.19, 0.02) | −0.08 (−0.19, 0.02) | −0.16** (−0.26, −0.06) | −0.16** (−0.26, −0.05) |
| OECD member | 0.00 (−3.38, 3.38) | −0.02 (−3.41, 3.38) | −2.66* (−5.09, −0.23) | −2.63* (−5.06, −0.21) | −1.35 (−4.18, 1.47) | −1.32 (−4.14, 1.50) |
| Population density | 19.43 (−9.55, 48.40) | 19.44 (−9.61, 48.50) | 1.11 (−15.48, 17.70) | 0.85 (−15.42, 17.11) | 4.74 (−11.34, 20.82) | 4.38 (−11.17, 19.94) |
| Urban residence | 0.74* (0.17, 1.31) | 0.74* (0.17, 1.31) | 0.58 (−0.01, 1.17) | 0.56 (−0.01, 1.14) | 0.31 (−0.11, 0.73) | 0.29 (−0.11, 0.69) |
| Age dependency ratio | −0.52*** (−0.73, −0.31) | −0.52*** (−0.73, −0.31) | 0.01 (−0.16, 0.19) | 0.01 (−0.16, 0.18) | 0.03 (−0.14, 0.19) | 0.02 (−0.14, 0.19) |
| Male education | 1.15 (−2.60, 4.91) | 1.07 (−2.67, 4.82) | 2.03 (−1.33, 5.39) | 2.08 (−1.25, 5.41) | 3.12 (−0.09, 6.32) | 3.19* (0.02, 6.36) |
| Temperature | −0.82* (−1.56, −0.08) | −0.83* (−1.57, −0.09) | −0.78 (−1.59, 0.03) | −0.76 (−1.56, 0.04) | −1.02* (−1.87, −0.17) | −1.00* (−1.83, −0.17) |
| Rainfall | 0.35 (−1.06, 1.76) | 0.35 (−1.06, 1.76) | −0.49 (−1.78, 0.80) | −0.47 (−1.76, 0.81) | 0.14 (−1.06, 1.35) | 0.17 (−1.03, 1.37) |
| Earthquakes | −0.98 (−2.12, 0.15) | −0.94 (−2.04, 0.17) | −0.80 (−1.72, 0.12) | −0.84 (−1.76, 0.07) | 0.29 (−0.54, 1.11) | 0.23 (−0.58, 1.04) |
| Droughts | −1.78* (−3.14, −0.42) | −1.80* (−3.17, −0.43) | 0.13 (−0.61, 0.87) | 0.11 (−0.63, 0.84) | −0.11 (−1.02, 0.79) | −0.15 (−1.04, 0.74) |
| Observations | 1,527 | 1,527 | 3,367 | 3,367 | 3,367 | 3,367 |
| Countries | 177 | 177 | 180 | 180 | 180 | 180 |

*$p < 0.05$,

**$p < 0.01$,

***$p < 0.001$. Data given as adjusted beta coefficient (95% confidence interval). Robust standard errors were employed. Each column is the output from 1 panel regression with fixed effects adjusted for the covariates in the table in addition to year dummies (not shown). Coefficients are interpreted as the absolute change in the dependent variable following a 1-unit change of the independent variable. GDP per capita is in current US dollars, and its unit is scaled up by 1,000. Population density represents the percentage of the population living in a density of >1,000 people/km$^2$. Urban residence represents the percentage of the population living in urban areas. The age dependency ratio represents the proportion of dependents (aged under 15 years or over 64 years) per 100 working-age individuals. Male education is expressed as years per capita and is age-standardised. Temperature is in degrees Celsius and is the mean population-weighted annual temperature. Rainfall is the mean population-weighted annual rainfall in millimetres per year, scaled down by 1,000. Earthquake and drought are binary variables representing their absence or presence. All armed conflict variables were lagged by 1 year.

GDP, gross domestic product; OECD, Organisation for Economic Co-operation and Development.

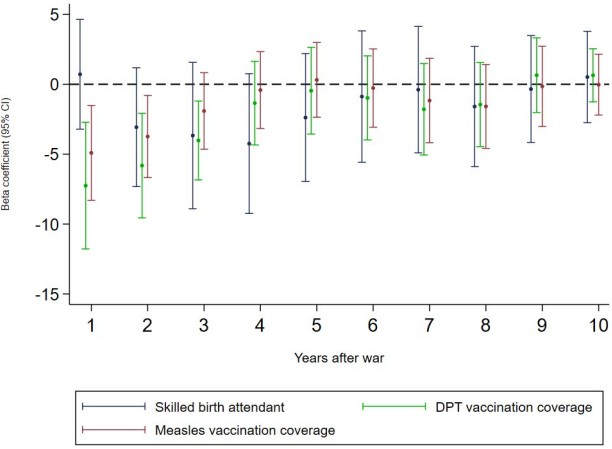

**Fig 2. The association between war and delivery by a skilled birth attendant and childhood vaccinations with up to 10-year lags.** Each line represents the output from a fixed effects panel regression model adjusted for 10 covariates and year fixed effects, showing the war coefficient only. DPT, diphtheria, pertussis, and tetanus.

geographies of previous work, taking an approach that adjusts for country-level factors and explores time trends while accounting for pre-conflict and post-conflict trends.

These associations exemplify the susceptibility of mothers and children during periods of conflict. They are likely driven by the known impacts of conflict, including reduced government spending on healthcare and damage to healthcare infrastructure (deliberate or collateral) [6,35]. Reduced utilisation, access, and affordability of high-quality maternal and child services [35,36] can directly affect individuals' risk of mortality. Additionally, changes to security, governance, health workforce, financing, supplies, and monitoring can affect the quality of maternal and child healthcare [37]. For example, interviews with government and UN officials working in conflict-affected Nigeria and Mali highlighted the difficulty in planning for healthcare resource allocation due to constant population movement, and in accessing remote populations in insecure settings [38,39], which explained poor maternal and child health indicators. Further empirical evidence from Nigeria has found that living in close proximity to development aid during conflict was associated with lower infant mortality [15].

An established literature links armed conflict to lower workforce retention and increased job stress and insecurity [40,41], but this study did not find significant associations between conflict and changes in skilled birth attendance. Reductions in workforce retention may be partly offset by increased humanitarian support, particularly in the short term. For example, in Syria, humanitarian agencies were able to successfully advocate funding for prioritisation of reproductive and maternal healthcare [42]. Research from Afghanistan showed that coverage of skilled birth attendants and institutional facility deliveries continued to increase linearly between 2003 (approximately 20%) and 2015 (approximately 50%), when the country experienced substantial national conflict, explained by early investment in maternal health programmes and a resilient health system [37]. A similar pattern was recently documented in Colombia [43]. The absence of a significant relationship between conflict and neonatal mortality indicates that mechanisms underlying the association between conflict and maternal and child health may be complex and may vary between outcome measures. One interpretation is that birth is a key event for which mothers are most likely to seek care, even in fragile and insecure environments, which, coupled with the skilled birth attendance, could mitigate against the risk of neonatal mortality. In contrast, conflict may dissuade mothers from seeking healthcare during pregnancy and infancy, explaining increases in maternal and infant mortality and

reduced vaccination uptake. This is supported by research conducted among Syrian refugees in Lebanon, which found that expectant mothers preferred to return to war-torn Syria for planned cesarean sections, to prevent against uncertainties in reaching a hospital during natural labour (e.g., blocked roads, night curfews) [44]. Furthermore, research from conflict-affected Burundi showed that integrating traditional birth attendants into the primary healthcare system as 'birth companions' encouraged expectant mothers to seek healthcare rather than give birth at home [45]. It is also plausible that our results for skilled birth attendants are affected by selection bias, given over half of the global data were missing for skilled birth attendance and missing data were more frequent in conflict-affected countries, where data availability and quality are limited.

The results on child vaccinations—which can act as a proxy for coverage of other key healthcare interventions—suggest that health services may be a key pathway through which conflict negatively affects maternal and child health. However, such effects are unlikely to be universal as studies of conflict-affected refugees have found improvements in maternal and child health indicators due to health services provided in refugee camps [46–48]. Populations that are unable to flee conflict may be the most affected by deteriorating health systems, but other non-health-service pathways likely play a role.

Previous limited evidence indicates that deteriorations in socioeconomic and environmental conditions may also contribute to increased maternal and child mortality during and after conflict [14]. Modelling studies from sub-Saharan Africa have found that conflict is associated with higher rates of child malnutrition [49,50]—a contributor to infant and child mortality. Armed conflict has been also shown to have a negative impact on school enrolment, with girls disproportionately affected [51]. Child labour may be another contributing factor, with girls found to be exposed to harsher working conditions than boys in conflict situations [52]. Other research has found associations between armed conflict and physical injuries, infectious diseases, poor mental health, and poor sexual and reproductive health in women and children [35].

This study highlights how the negative health outcomes associated with conflict may appear and persist several years after the onset of war, particularly with respect to under-5 mortality. Delayed associations between armed conflict and health outcomes may reflect an amalgamation of effects on several upstream and healthcare factors, such as the withdrawal of humanitarian funding, the slow breakdown of vital health and social infrastructure, and the accumulation of risk factors in pre-birth and in early life, including delays to immunisation, increased diarrhoea, and the absence of antibiotics [53].

This study is not without limitations. The analyses are ecological, precluding causal inference at the individual level. A country-level analysis likely leads to conservative point estimates for conflict, as data reflect country averages and mask sub-national variation in conflict intensity and likely differential associations across socioeconomic groups [54]. Studies using individual-level health data from Demographic and Health Surveys (DHS) seek to address this limitation by better capturing conflict activity in local geographies [10–13]. However, the UN Maternal Mortality Estimation Inter-agency Group cautions against using DHS data for monitoring trends as the program's methodology (asking respondents about survivorship of sisters) generates estimates pertaining to any time within the past few years before the survey [55]. A key strength of our study is therefore the use of data from the World Bank, which unlike DHS, enables an assessment of trends and the associations of shocks such as armed conflict with maternal mortality.

This study may also be limited by the data sources; for example, there may be biases introduced by estimation and data processing by data sources before publication (e.g., WHO or Institute for Health Metrics and Evaluation), which may mask sudden shifts in mortality

associated with conflict and produce conservative estimates. However, we have shown previously that such data processing did not obscure abrupt changes in mortality in conflict-affected countries and are unlikely to introduce substantial bias to this analysis [19]. In terms of outcomes, this study only examined maternal and child mortality, and selective health service indicators. More detail by causes of death in addition to other health systems data could perhaps better elucidate underlying patterns and mechanisms of action, including our lack of association between armed conflict and neonatal mortality. The analytical approaches employed in this study are robust and frequently used in global country-level analyses, and the use of fixed effects specifications aims to remove biases from country-level characteristics. Nonetheless, there remains the potential for unobserved biases that could explain these findings despite our employing multiple confounders and robust model specifications. The role of conflict duration was not incorporated into the analytical approach, and this should be considered as an additional dimension in assessing conflict impacts in future work.

This study shines light on the major challenges that countries in conflict face in meeting Sustainable Development Goal targets for maternal and child health, especially in settings where conflict is of high intensity. Beyond the recommendation that all conflicts should cease immediately, secondary prevention measures should focus on mitigating the effects of armed conflict on maternal and child health. Our results demonstrate the importance of protecting health system functioning during periods of conflict. Additionally multi-sectoral humanitarian and development approaches are required, including attention on food security, economic development, poverty alleviation, war prevention advocacy, and improved education. It is imperative that international efforts to support conflict-affected countries are integrated with existing infrastructure to enable long-term and sustainable improvements in health systems and socioeconomic and environmental conditions once acute deteriorations have been addressed [56]. Combined, these actions are vitally important to protect maternal and child health during and after periods of conflict.

## Supporting information

**S1 RECORD Checklist. The RECORD statement—checklist of items, extended from the STROBE statement, that should be reported in observational studies using routinely collected health data.**
(DOCX)

**S1 Table. Definition of covariates.**
(DOCX)

**S2 Table. The association between armed conflict and maternal and child mortality (beta coefficients, 95% confidence intervals): Additional conflict specifications.**
(DOCX)

**S3 Table. The association between armed conflict and maternal and child health indicators (beta coefficients, 95% confidence intervals): Additional conflict specifications.**
(DOCX)

**S4 Table. Sensitivity analysis: Random effects specification (maternal mortality).**
(DOCX)

**S5 Table. Sensitivity analysis: Sequential addition of covariates, including a complete case analysis (maternal mortality).**
(DOCX)

**S6 Table. Sensitivity analysis: Major Episodes of Political Violence exposure data (maternal mortality).**
(DOCX)

## Acknowledgments

We would like to thank Dr. Jasper Been for providing helpful comments on the manuscript.

The views expressed in this paper are those of the authors and do not necessarily reflect the views of the United Nations.

## Author Contributions

**Conceptualization:** Mohammed Jawad, Thomas Hone, Eszter P. Vamos, Valeria Cetorelli, Christopher Millett.

**Data curation:** Mohammed Jawad, Thomas Hone.

**Formal analysis:** Mohammed Jawad, Thomas Hone.

**Methodology:** Mohammed Jawad.

**Supervision:** Thomas Hone, Eszter P. Vamos, Christopher Millett.

**Writing – original draft:** Mohammed Jawad, Thomas Hone, Christopher Millett.

**Writing – review & editing:** Mohammed Jawad, Thomas Hone, Eszter P. Vamos, Valeria Cetorelli, Christopher Millett.

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
