## [Editor Report · Decision Letter 0]

16 Feb 2021

Dear Dr Jawad, 

Thank you for submitting your manuscript entitled "The association between armed conflict and maternal and child health: a panel regression analysis of 180 countries, 2000-2019" for consideration by PLOS Medicine for our upcoming Special Issue.

Your manuscript has now been evaluated by the PLOS Medicine editorial staff as well as by the Guest Editors and I am writing to let you know that we would like to send your submission out for external assessment.

Once your full submission is complete, your paper will undergo a series of checks in preparation for external assessment. 

Kind regards,

Richard Turner, PhD

rturner@plos.org

---

## [Decision Letter · Decision Letter 1]

26 Apr 2021

Dear Dr. Jawad,

Thank you very much for submitting your manuscript "The association between armed conflict and maternal and child health: a panel regression analysis of 180 countries, 2000-2019" (PMEDICINE-D-21-00734R1) for consideration at PLOS Medicine for our upcoming Special Issue. 

Your paper was evaluated by the Guest Editors and sent to independent reviewers, including a statistical reviewer. The reviews are appended at the bottom of this email and any accompanying reviewer attachments can be seen via the link below:

[LINK]

In light of these reviews, we will not be able to accept the manuscript for publication in the journal in its current form, but we would like to invite you to submit a revised version that addresses the reviewers' and editors' comments fully. You will appreciate that we cannot make a decision about publication until we have seen the revised manuscript and your response, and we expect to seek re-review by one or more of the reviewers. 

We hope to receive your revised manuscript by May 17 2021 11:59PM. Please email us (plosmedicine@plos.org) if you have any questions or concerns.

Please let me know if you have any questions, and we look forward to receiving your revised manuscript. 

Sincerely,

Richard Turner, PhD

rturner@plos.org

Please adapt the title to better match journal style. We suggest: "Implications of armed conflict for maternal and child health: A regression analysis of data for 180 countries from 2000-2019".

Please add a new final sentence to the "Methods and findings" subsection of your abstract, which should begin "Study limitations include ..." or similar and should quote 2-3 of the study's main limitations. 

At line 43, please adapt the sentence to begin "Our analysis indicates that ..." or similar.

After the abstract, please add a new and accessible "Author summary" section in non-identical prose. You may find it helpful to consult one or two recent research papers published in PLOS Medicine to get a sense of the preferred style. 

At line 53, please make that "... 5 years".

Early in the methods section of your main text, please state whether the study had a protocol or prespecified analysis plan, and if so attach the relevant document(s) as a supplementary file(s), referred to in the text. 

Please highlight analyses that were not prespecified. 

At line 96 and any other instances, please make that "data ... were".

Throughout the text, please format reference call-outs as follows: "... child malnutrition [34,35] ..." (i.e., no spaces in the square brackets).

Throughout the paper, please use the style "up to 2 years", although numbers should be written out at the start of sentences. 

Please remove all instances of "[online]" from the reference list.

Noting reference 13, please ensure that all references have full access details. 

Please update reference 41, and if not published make the latest version available to us in confidence. 

Please add a completed checklist for the most appropriate reporting guideline, which may be RECORD, labelled "S1_RECORD_Checklist" or similar and referred to as such in the Methods section. 

In the checklist, individual items should be referred to by section (e.g., "Methods") and paragraph numbers, not line or page numbers as the latter generally change in the event of publication. 

Comments from the reviewers:

*** Reviewer #1: 

This study aims to assess the association between conflict and maternal and child health globally, in both the short and long term.

Comments:

There are just a few typos to correct within the manuscript.

"Due to the complexity of measuring conflict, four specifications were explored in the analyses."

The authors have demonstrated rigor by applying four different specifications here, including the number of battles being considered as a continuous variable.

"To adjust regression models for potential confounders, covariates were selected based on conceptual frameworks that outlined the theoretical causal pathways between conflict and health [25-27]. Ten suitable covariates were identified that are probable confounders and included in the regression models - all expressed at the country-year level. "

These are: "GDP per capita, OECD member, population density, urban living, age dependency ratio, male education, temperature, rainfall, earthquakes, and droughts"

The authors have suitably assessed the evidence base and attempted to account for various confounders within the models.

Additionally, did the authors consider a measure of accessibility to healthcare or healthcare systems as a potential confounder?

"There was missing data in four variables: GDP per capita (2.7%), age dependency ratio (3.9%), Multiplicative Polyarchy Index (11.8%), and Ethnic Fractionalisation Index (46.0%). The latter two were omitted from models due to high levels of missing data, but reintroduced in sensitivity analyses."

The authors have appropriately assessed and accounted for missing data within their analysis, including running a sensitivity analysis on this.

Did the authors consider completing a multiple imputation methodology for missing data for comparison, to further assess the robustness of their findings?

Overall, this study presents a clearly described methodology, where collinearity and outliers are adequately explored and removed from the model, and a thorough set of sensitivity analyses have been completed.

Furthermore, the study limitations have been suitably and transparently discussed.

*** Reviewer #2: 

General Comments: This is an interesting and fairly straightforward paper on an important topic that is gaining much attention. While other recent papers have used the Uppsala Conflict Database to examine similar questions, this study takes a different modeling approach using different datasets. Further, instead of focusing at the sub-national level, they conduct an ecological analysis at the country level, which has some strengths and weaknesses. While the paper in general is clearly written, there are several areas where the authors could improve. Notably, the discussion section presents an overly simplistic picture of the findings and focuses primarily explaining the results they observe as stemming from decreases in government spending. Further, they provide little reflection or interpretation of the results, rather the discussion repeats much of the same conceptual rationale for why conflict is associated with increased maternal, child, and infant mortality. 

Specific comments:

* Line 137: The authors made a decision to not include countries that have many minor conflicts summing to more than 1000 battle-related deaths per year with the countries that have experienced war. It would be helpful it the authors could provide more information as to what they mean by multiple minor conflicts and to provide a rationale for this distinction. If there are many minor conflicts in a country that amount to the same number of deaths, why would this not have a similar impact as a war on the health system or macro-level environment, thus leading to the same outcomes as hypothesized resulting from exposure to war? Did the authors perform any sensitivity analysis to interrogate this decision?

* In the methods, some of the co-variates are not fully defined. For example, the authors do not define how skilled birth attendance is measured. Authors should make sure that all exposures and covariates are clearly defined. 

* As stated above, the discussion section presents a rather simplistic interpretation of the results and focuses primarily on reduced government spending on healthcare and damage to healthcare infrastructure. However, there are several other factors that could contribute to the increases observed and be equally important. For example, conflict likely limits a woman's ability to travel to a facility for ANC and delivery; though of note, it is interesting that the authors do not find a significant decrease in skilled birth attendance associated with conflict. To my earlier point, a clearer definition of what skilled birth attendance means in this study would be helpful to develop a more nuanced picture of the results. Are the authors able to further unpack skilled birth attendance by looking at institutional births or cadre of "skilled" attendant? The authors do state that this finding could be the result of a type II error, though it is noted that the confidence intervals around the point estimates are not particularly wide. Providing some deeper discussion here could add depth to the interpretation of these results. 

* Further, the authors do not reflect on some of their more interesting findings in the discussion. For example, it is quite interesting that under-5 mortality increases only between three and eight years since the onset of war. Why might this be the case and why might the effects not be immediately observed?

* Similarly, I think one of the more important findings of this paper is the significant lagged effects observed with regard to several of the outcome variables. While the authors reference this finding in their conclusions; they give little interpretation of these results in the discussion section. 

* Given that other studies have recently looked at maternal mortality (several of which are cited), and other related indicators, such as institutional delivery, using the same database to assess conflict exposure, it would be helpful if the authors could contextualize the results of this study in the existing literature. 

* The authors mention in the limitations that "DHS does not provide a representative picture of conflict exposure and so can over- or underestimate the effects of conflict depending on the sampling approach [39]. Additionally, DHS analyses are limited in their ability to assess trends and impact of shocks such as conflict [40]- which is a key strength of this study." These statements are not clear. In particular, DHS does not measure conflict and is not conducted during times of conflict, thus it is not clear how the DHS would under or over-estimate the effects of conflict. Further, it is not clear what the authors are referring to their improved ability over these other studies to "assess trends and shocks such as conflict," given that the data used to measure conflict is this analysis is the same as that used in the analyses that use DHS. More clarity here is needed. The papers referenced by the authors Further, these statements are not appropriately referenced. Reference 39 is not directly relevant to the DHS while reference 40 appears to have been made in error. 

* Given that several other papers have examined conflict exposure using smaller scale geographies, and this paper takes a country-level ecological approach, it would be helpful to see a more nuanced discussion of why this approach is particularly meaningful, what new can be learned from it beyond what has already been published, etc. beyond the little that is mentioned in the limitations section. 

* Page 21 (no line numbers): what do the authors mean by "data manipulation?" This is not clear.

* In conclusion, the authors state "This study shines light on the major challenges that countries in conflict face in meeting SDG targets for maternal and child health, especially in settings where conflict is of high intensity and long lasting." I do not see where the authors examine interactions between duration and intensity of conflict. It would seem that this would be a particularly salient dimension to explore—especially given that the effect of prolonged conflict on systems-level deterioration may compound over time as opposed to conflicts that are more brief in duration. 

*** Reviewer #3: 

This is an interesting and relevant study that assesses the association between conflict and maternal and child health, analyzing six indicators in 180 countries between 2000 and 2019 with data from the Uppsala Conflict Data Program and World Bank. The statistical methods are adequate and robust to draw the conclusions. 

Some minor comments that may help to improve the manuscript are:

Introduction: I suggest to add a paragraph that introduce some of the other variables associated with the outcomes of interest that are taken into account as potential confounders in the analysis and how some of this are highly correlated with conflict. 

Methods: 

I suggest expanding the explanation of the population mean temperature and rainfall, it is not very clear. 

Page 9, statistical analysis: "Four data points (Rwanda 1994, Bosnia and Herzegovina 183 1995, Congo 1997, and Eritrea 1999) were considered outliers due to having high leverage 184 and influence, and were omitted from the model.." please expand or clarify exactly to what variable are you referring. 

Results: Table 2: improve the title to include that it is an adjusted model. 

It seems that there is a mistake in the number of the figure captions. 

Is there a way to account for possible correlation or collinearity between conflict and confounding variables?

I suggest including a brief description of the findings related to the confounding variables in the models.

Discussion:

The Branch consortium has a series of recent publications that could help to expand the comparisons with other studies. 

https://www.biomedcentral.com/collections/branchconsortium

***

[LINK]

---

## [Decision Letter · Decision Letter 2]

10 Jul 2021

Dear Dr. Jawad,

Thank you very much for submitting your revised manuscript "Implications of armed conflict for maternal and child health: A regression analysis of data for 180 countries from 2000-2019" (PMEDICINE-D-21-00734R2) for consideration at PLOS Medicine for our upcoming Special Issue. 

Your paper was discussed with the guest editors for the issue, and re-seen by our independent reviewers, including a statistical reviewer. The reviews are appended at the bottom of this email and any accompanying reviewer attachments can be seen via the link below:

[LINK]

In light of these reviews, we will be unable to accept the manuscript for publication in the journal in its current form, but we would like to invite you to submit a further revised version that addresses the reviewers' and editors' comments fully. We will not be able to make a decision about publication until we have seen the revised manuscript and your response, and we expect to seek re-review by one or more of the reviewers. 

We hope to receive your revised manuscript within two weeks. Please email us (plosmedicine@plos.org) if you have any questions or concerns.

Please let me know if you have any questions, and we look forward to receiving your revised manuscript. 

Sincerely,

Richard Turner, PhD

rturner@plos.org

Please address the issues of selection bias and missing data for each of the main outcomes (with summary data on each provided for review)

We ask you to add discussion to put your findings in context of other related results from the BRANCH and PRIO consortia.

We also suggest devoting some discussion to the findings on deliveries with skilled birth attendants and neonatal mortality, where the observed impacts may be less than would be expected. 

Please adapt the data statement (submission form) to provide the links where study data can be obtained. 

Thank you for describing the origin of your analysis plan. Are you able to attach a document or documents as a supplementary file, referred to in the methods section?

Please use the journal name abbreviation "PLoS ONE" in the reference list; and use other abbreviations, e.g., "Confl Health", consistently.

Comments from the reviewers:

*** Reviewer #1: 

The authors have satisfactorily considered and responded to each comment in turn.

*** Reviewer #2: 

The authors have generally done a good job addressing my previous comments; however, there are still some issues that need to be addressed that have emerged as a result of some of the revisions. 

1. The authors mention that over half of the countries are missing data on SBAs and that more conflict-affected countries are missing data on this variable than non-conflict affected countries. This seems like a substantial limitation in the paper and it is difficult for me to understand how this offers a robust estimate of the association given the extent of missing data, but the authors only make a passing comment about this issue. Are any of the other variables of interest missing data? It would be helpful for the authors to provide this additional information for other important variables in the paper that may also be subject to selection bias in this same way due to missing data. 

2. The authors discuss the limitation with the DHS data as it being unable to assess trends. "A key strength of our study is therefore the use of data from the World Bank, which unlike the DHS, enables an assessment of trends and the impacts of shocks such as armed conflict." However, the limitation that they describe really only seems to be relevant to estimates of maternal mortality given the DHS use of the sisterhood method, and it is hard to understand how this would apply to other other outcomes of interest investigated by the authors. The authors should be more precise in discussing these limitations. 

*** Reviewer #3: 

The authors have addressed the comments satisfactorily.

***

[LINK]

---

## [Decision Letter · Decision Letter 3]

2 Sep 2021

Dear Dr. Jawad,

Thank you very much for re-submitting your manuscript "Implications of armed conflict for maternal and child health: A regression analysis of data for 180 countries from 2000-2019" (PMEDICINE-D-21-00734R3) for consideration at PLOS Medicine for our upcoming Special Issue.

I have discussed the paper with editorial colleagues and the guest editors, and it was also seen again by one reviewer. I am pleased to tell you that, provided the remaining editorial and production issues are fully dealt with, we expect to be able to accept the paper for publication in the journal.

[LINK]

Please let me know if you have any questions, and we look forward to receiving the revised manuscript.   

Sincerely,

Richard Turner, PhD

rturner@plos.org

Requests from Editors:

At line 31, prior to the presentation of the findings, we suggest adding a sentence to summarize the number of observations, number of minor conflicts and number of wars, for example. 

Parenthetically, why are 192 countries specified in table 1 when you highlight 180 countries in the title etc?

At line 36 and elsewhere in the paper, please avoid the word "effects" given the observational study design, substituting "impacts" or similar. 

At line 36 we suggest amending the wording to: "Long term impacts of war were demonstrated by associated increases in maternal mortality observed for up to 7 years; in under 5 mortality ..." or similar.

At line 46, "... relating to"?

Please spell out "DPT" at first use in abstract, author summary and main text.

Comments from Reviewers:

Reviewer #2: 

The authors have addressed all my concerns. This is an interesting paper that will make an important contribution.

***

[LINK]

---

## [Editor Report · Decision Letter 4]

13 Sep 2021

Dear Dr Jawad, 

On behalf of my colleagues and the Academic Editor, Dr Bhutta, I am pleased to inform you that we have agreed to publish your manuscript "Implications of armed conflict for maternal and child health: A regression analysis of data for 180 countries from 2000-2019" (PMEDICINE-D-21-00734R4) in PLOS Medicine.

Prior to final acceptance, please:

Around line 126 (Methods), adapt the text to "Although a written plan is not available, the analysis plan was based ..." or similar (assuming this is correct); 

Adapt the text to "... continued to increase linearly" (soon after the call-out for reference 42);

And remove "forthcoming" from reference 25 (if this is available online, please include a URL with accessed date).

PRESS

Sincerely, 

Richard Turner, PhD 

rturner@plos.org